# Reflections Based on Pollution Changes Brought by COVID-19 Lockdown in Shanghai

**DOI:** 10.3390/ijerph182010613

**Published:** 2021-10-10

**Authors:** Fang Zhang, Yi Han, Bailin Cong

**Affiliations:** 1MNR Key Laboratory for Polar Science, Polar Research Institute of China, Shanghai 200136, China; 2Department of Earth Sciences, Durham University, Durham DH1 3LE, UK; yi.han2@durham.ac.uk; 3First Institute of Oceanography, Ministry of Natural Resources (MNR), Qingdao 266061, China

**Keywords:** Shanghai, different functional areas, major airborne pollutants, COVID-19, effective policy, sustainable societies

## Abstract

COVID-19 and its variants have been changing the world. The spread of variants brings severe effects to the global economy and to human’s lives and health, as well as to society. Lockdown is proven to be effective in stopping the spread. It also provides a chance to study natural environmental changes with humanity’s limited interference. This paper aims to evaluate the impact of lockdown on five major airborne pollutants, i.e., NO_2_, SO_2_, O_3_, PM_2.5_ and PM_10_, in the three different functional regions of Chongming, Xuhui and Jinshan of Shanghai. Changes in the same pollutants from the three regions over the same/different periods were all studied and compared. Overall, the COVID-19 lockdown has changed pollutant concentrations in the long and short terms. Concentrations of four pollutants decreased, except for that of earth surface O_3_, which increased. SO_2_ had significant correlations with all other pollutants. PM_2.5_ and PM_10_ are more externally input than locally produced. NO_2_, SO_2_ and PM levels sharply reduced in Jinshan and Xuhui due to the limited usage of fossil fuel. Lockdown improved the air quality. People now have a chance to rethink the value of life and the harmony between economic progress and environmental protection. This is helpful to establish sustainable societies.

## 1. Introduction

Although more than one and half years has passed since the first case of infection was reported in Wuhan, more and more variants of COVID-19 have been found, such as *α*-, *ß*-, *γ*-, *δ*-, *ε*-, *λ*- and *μ*-strains, leading to more than 2.36 billion and 4.66 million confirmed and dead cases, respectively, as of 15 September 2021 [1]. *δ*(plus) -, *λ*- and *μ*-strains are three newfound variants [2]. The spread and infection rates of these are more serious than the original. They are putting the world in a more difficult and dangerous state by causing more deaths and severe cases. The *δ*-variant was first found in India on 11 June 2020, and has been found in more than 130 countries since. It has put Europe back on “war footing” [3]. The *δ*-variant is highly contagious, with a higher speed of infection and a short latent period. The proportion of the *δ*-variant is greater and greater among COVID-19 cases [4], and can induce one time more deaths than other variants [5]. This has a serious impact on the world economy and human health [3,4,5,6]. The virus is a serious blow to social stability [6,7,8]. The first infection of COVID-19 was detected in Wuhan in December 2019 [2], and China responded quickly to stop its spread by starting the first-level response in January 2020 [6,7,8]. The responses included traffic restrictions and factory closures [5,6,7]. The reported peak number of cases occurred on 12 February [7], after which the cases gradually decreased to single digits [7].

The development of human society relies too much on traditional energy sources, such as coal and oil, which produce large quantities of polluting gases. Therefore, air pollution is a major problem at present, which has serious impacts on climate change and human health. We need to know the basic changes that occur in levels of pollution gases when human intervention is minimized, so as to compare with the pollution gases produced by the normal operations of human society and thus obtain the basic data of human intervention. The COVID-19 lockdown gave humanity the perfect chance [9,10,11]. Different regions in the world had different polluting gasses [9]. The major air pollutants are NO_2_, SO_2_, O_3_, PM_2.5_ and PM_10_ in Shanghai [10,12]. NO_2_ produces photochemical smog and is oxidized by O_3_ to produce NO_3._ This causes photochemical smog and acid rain [12,13,14,15]. Both traffic and heavy industry produce SO_2_ [14,15,16,17]. Short-term exposure to SO_2_ can increase death rates [18]. Surface O_3_ usually forms in agricultural and forestry areas around large cities [14,16,17]. Its concentration varies with the season and with meteorology [14,19,20]. High O_3_ concentration damages human health and agriculture production [8,14,19,20,21], because O_3_ oxides nitrogen in the presence of VOCs and light [14,16,17,22,23,24]. Both PM_2.5_ and PM_10_ are harmful to human health, as some toxic gases and pathogenic microorganisms, such as bacteria and viruses, may attach to them [13,15,19,20,21,22,23,24,25,26]. PM_2.5_ has caused more than 2.1 million people’s deaths because of its acidity penetrating the lower airways [13,24,26,27,28,29]. Short term exposure to these pollutants will increase the risk of COVID-19 infection [13,18]. Daily average concentrations of the five pollutants were collected from 1 January to 9 February 2020 [12]. Data of the same pollutant belonging to different functional areas were compared to make a comprehensive study. In addition, the corresponding average data from the years 2018 and 2019 were used as a baseline to make a comparison.

There have been a lot of studies on the environmental changes during the breakout and lockdown of COVID-19 [1,10,13,15,27,30,31,32,33,34,35,36]. However, most of these studies are concentrated during the lockdown [10,11,19,27,31,32,33,34,36,37,38,39,40,41,42] without a comparison between pre- and post-lockdown numbers. In addition, seldom do studies refer to functional areas [10,20]. These two issues were specifically focused on by our study. COVID-19 and its variants have altered the world in many respects, including human lives and their physical and mental health [1,2,20,27,35]. However, it also provides a special chance for humans to rethink the value of life, cherish their lives and jobs, make appropriate measures to balance economic growth with a clean natural environment, and steady the progress of the economy [11]. Thus, although most effects brought about by the variants are negative, they give mankind the chance to restart the world and establish sustainable societies.

## 2. Methods

### 2.1. Data Collection

The daily data of the pollutants (NO_2_, SO_2_, O_3_, PM_2.5_ and PM_10_) were provided by the Shanghai Municipal Bureau of Ecology and Environment (https://sthj.sh.gov.cn/, accessed on 1 January–23 January 2020), and daily average concentrations were used. Three regions in Shanghai (Figure 1), i.e., Chongming, Xuhui and Jingshan stations, were chosen as different functional areas for data analysis. Chongming represents the area covered by large amounts of vegetation. Its total area is 50.94 km^2^ and population is 678,000, with a density of about 1 person per km^−2^ by the end of 2018. Xuhui is the Central Business District (CBD) area in Shanghai. Its total area is 1413 km^2^ and it had 818,100 permanent residents [10], with a density of about 13,310 persons per km^−2^, by the end of 2018. Xuhui is the typical area to study how concentrations of the major air pollutants vary in a populated urban environment. There are 176 companies registered in the Jinshan industrial zone and most of them are chemical companies. In addition, 59 of those firms emit pollutants [10]. Its total area is 50.94 km^2^ and the registered population was 522,300 with a density of about 10,253 persons per km^−2^ by the end of 2020. We collected an environmental dataset containing 613 values during 2018–2020 from all three stations.

### 2.2. Data Processing

Two comparisons were done: First, a comparison between before (1 January–23 January) and during (24 January–9 February) lockdown in the year of 2020. Second, a comparison between data from the same dates but of different years (average data of 2018–2019 and data of 2020). One-way ANOVA, Nonmetric Multidimensional Scaling (NMDS) analysis and cluster analysis were used to process the data. SPSS (SPSS Inc., Chicago, IL, USA) was used to do one-way ANOVA, whereas R was used to do cluster analysis (version 3.4.1). The calculations for the Air Quality Sub-Index (AQI) and Air Pollutant (IAQIP) are as follows: (1)$$AQI=\max\left(I_{1},I_{2},\ldots ,L_{n}\right);\;\;IAQI_{P}=\frac{I_{high}-I_{low}}{C_{high}-C_{low}}\left(C_{P}-C_{low}\right)+I{\;}_{low}$$
$$C_{P}\text{:}\;\mathrm{the}\;\mathrm{concentration}\;\mathrm{of}\;\mathrm{pollutant}\;p;\;C_{low}\text{:}\;\mathrm{the}\;\mathrm{concentration}\;\mathrm{breakpoint}\;\mathrm{that}\;\mathrm{is}\;\text{≤}\;C_{P}$$
$$C_{high}\text{:}\;\mathrm{the}\;\mathrm{concentration}\;\mathrm{breakpoint}\;\mathrm{that}\;\mathrm{is}\;\text{≥}\;C_{P};\;I{\;}_{low}\text{:}\;\mathrm{the}\;\mathrm{index}\;\mathrm{breakpoint}\;\mathrm{corresponding}\;\mathrm{to}\;C_{low};\;I{\;}_{high\;}\text{:}\;\mathrm{the}\;\mathrm{index}\;\mathrm{breakpoint}\;\mathrm{corresponding}\;\mathrm{to}\;C_{high}.$$

One-way ANOVA is used to compare the concentrations of the same pollutants between different groups, including before and during lockdown in the year 2020, and between the years of 2018–2019 and 2020. NMDS is a data analysis method that simplifies the research objects (samples or variables) in multidimensional space to low-dimensional space for positioning, analysis and classification, while preserving the original relationship between objects. According to the pollutant information contained in the samples, it is reflected in the multidimensional space in the form of points, and the degree of difference between different samples is reflected by the distance between points, and, finally, the spatial location map of the samples is obtained. Cluster analysis is a method to simplify data through data modeling. It is a set of statistical analysis techniques that divide research objects into relatively homogeneous groups. Clustering analysis is also called classification analysis or numerical classification. Cluster analysis can start from the sample data to automatic classification. If the variables are independent of each other, it can quickly process large datasets and automatically determine the optimal classification number of these classifications and continuous variables.

## 3. Results

### 3.1. Major Pollutants in Different Areas of Shanghai during COVID-19 in Year 2020 

Although concentrations of different pollutants had different maximum and minimum values in different parts of Shanghai, their changing trends were generally the same (Figure 2). Four of the five pollutants had decreasing concentrations while O_3_ had increasing concentrations, showing its maximum value at Chongming, which has a large number of green plants, followed by Xuhui, with a high population density, and Jinshan, with heavy industry. This was in line with the oxidation effect of O_3_ to NO_x_ in the presence of VOCs [12,23,43]. Hence, very significant negative correlations (*p* < 0.001) were shown between NO_2_ and O_3_ (Figure 3). NO_2_ decreased immediately after the lockdown started, as it is mainly produced by cars, with maximum values in Xuhui which has the greatest population density and the most cars. Correlations between PM and O_3_ were also negative, because some VOCs and NO_x_ were attached to PM [23,27,43]. Due to the Spring Festival being celebrated with family barbecues and the reopening of some factories, little peaks appeared after about seven days during the later period of the Chinese Spring Festival (Figure 3). The celebration increased NO_x_ and VOCs, which increased the concentration of O_3_ [10,23,27,43]. The significant (*p* < 0.05) positive correlation between PM_2.5_ and O_3_ before lockdown shows that the increase in PM_2.5_ attaching with NO_x_ and VOCs also increased the concentration of O_3_.

### 3.2. Comparison of Major Pollutants in Different Functional Areas

In Chongming (Figure 3), the concentrations of both NO_2_ and SO_2_ decreased by 51%, which was less than half of their original level during the blockade. Comparatively, the concentration of PM_2.5_ only decreased by 20% of its concentration before the blockade. The average concentration of O_3_ almost maintained its value, with 79.4 and 79.1 μg/m^3^, respectively, before and after the blockade. In Xuhui, the concentration of NO_2_ and SO_2_ decreased by 24% and 28%, respectively, and PM_10_ and PM_2.5_ decreased by 28% and 14%, respectively. However, the concentration of O_3_ increased by 9% during the blockade period. Comparatively, the concentration of PM_2.5_ only decreased by 20% compared to its concentration before the blockade. In Jinshan, NO_2_, SO_2_ and PM_10_ relatively decreased by 29%, 37% and 39% compared with the numbers from before the lockdown period. This is in accordance with this area having the most factories. Like Xuhui, the concentration of O_3_ also increased.

No significant effects on either PM_2.5_ or O_3_ were shown by the blockade in all the three parts of Shanghai (Figure 3), whereas very significant effects (*p* < 0.001) on NO_2_ and SO_2_ were shown. Weak (*p* < 0.05), significant (*p* < 0.01) and very significant (*p* < 0.001) effects were respectively shown on PM_10_ in the Chongming, Xuhui and Jinshan areas. This is in line with decreasing use of the two main resources of PM_10_, i.e., fossil fuel combustion and vehicle traffic. The significant decrease in NO_2_, SO_2_ and other reductants decreased O_3_ in both Xuhui and Jinshan. However, the results of PM_2.5_ were interesting, in that although the concentration also obviously decreased, no significant changes were found before and during the lockdown period in different functional parts of Shanghai. This indicates that local combustion of fossil fuels may not be the main source of PM_2.5_ [26,40]. Generally, all the pollutants had very significant (*p* < 0.001) reductions during the blockade. Only O_3_ increased by more than 20% of its levels before the lockdown. Comparatively, the concentration of SO_2_ decreased to 27%, and those of NO_2_, PM_2.5_ and PM_10_, decreased to 36%, 72% and 34%, respectively. This effect could last if proper polices are carried out, as shown by the gradually decreasing AIQ before, during and after the lockdown periods with their respective values of 85.7, 62.8 and 53.5. Most pollutants were clearly separated before and during the lockdown period (Appendix A); the same pollutants may have different sources in different functional areas (Appendix A).

### 3.3. Comparison of Major Pollutants on the Same Days from Different Years

All pollutants varied dramatically in 2020 compared with the same periods in 2018 and 2019, especially during the COVID-19 lockdown period (Figure 4). The average decrease of SO_2_, NO_2_, PM_10_, PM_2.5_ and O_3_ was by 46%, 10%, 32%, 34% and 3%, respectively, before the lockdown period, whereas the corresponding data were 46%, 54%, 38%, 34% and −16.2%, respectively, during the lockdown period. Therefore, the lockdown policy indeed altered the pollutants’ concentrations. Reducing heavy industry and motor vehicles are effective ways to control most pollutants.

## 4. Discussion

Exposure to air pollution may play an important role in COVID-19 incidence and deaths [1,4,13,23,24]. Especially NO_2_ and PM_2.5_, which are released from tailpipes [32], increase the risk of lung infections [10,12,14,15,30,31,32]. Among the five major pollutants in Shanghai, NO_2_ and SO_2_ are environmental indicators directly related to local economic activities [10]. More sustainable industry should be established for desulphurisation and to maintain SO_2_ at a low concentration [10,14]. NO_2_ concentration was at serious to pollution levels in Shanghai [6]. When the GDP in Shanghai reached 38,155.32 billion yuan in 2019 (China Statistics Bureau, 2019), NO_2_ became the only major pollutant that failed to meet the national standard (GB3095-2012). Besides of its role in causing lung disease, NO_2_ might be an important trigger of mental disorders [33,34] and is associated with morbidity and mortality of COVID-19 [35]. Lockdown in 2020 sharply decreased the concentration of NO_2_ by about 45% compared with the relative value before that period. All the data were about 28% and 46%, respectively, lower than the relative decreasing rates in Delhi and Mumbai [5], and about 20% lower in Almaty [35,36]. The global decreasing rates were about 9% and 10%, respectively, in March and April 2020 [5]. This was much higher than the decreasing rate during the lockdown period in Shanghai, but almost the same with that before the lockdown, i.e., from 1 to 23 January 2020 (Figure 2). Consequently, Shanghai performed well in controlling SO_2_ compared with other cities in the world_._ Comparatively, the decreasing rate of SO_2_ varied little before and during the lockdown periods in this study, but was about 10% higher than those released by traffic in 2020 [5]. Comparatively, the decreasing rates were 27% and 9% higher than those in Delhi and Mumbai. Consequently, SO_2_ levels were mainly from heavy industry and traffic, both of which use fossil as fuel. SO_2_ was the most decreased major pollutant in the year 2020 compared with its levels from 2018 and 2019. This is different from the data in Lima, Madrid, Moscow, Rome, Sao Paulo and Wuhan, where SO_2_ concentrations remained unchanged during lockdown because power plants and traffic were still operational [36,37]. 

Driven by particular meteorological conditions, PM is a primary pollutant during winter in Shanghai [12]. The wind contains pollutants coming across the Yangzi Delta seasonally from November onwards [5]. The concentrations of PM_2.5_ and PM_10_ result in more dust pollution [38,39]. The average concentrations of these two matters have dropped by 66% and 65% during the past five years due to proper control of gas emissions from factories in Shanghai and nearby [12]. The decrease of PM_2.5_ during the lockdown period was more than four times the average all over China (72%:17%) [40]. The decrease was about 10% lower than the values caused by traffic in the same periods in Shanghai (20~40%: 24–47%) [5], but still more than 10% higher than the corresponding values in Delhi and Mumbai of India [5]. Therefore, like SO_2_, the emission of PM_2.5_ is also well controlled in Shanghai compared with other international cities. There were almost no differences between the concentrations from the years 2018–2019 and 2020 in Shanghai before the blockade, which did decrease the concentration of PM_2.5_ by 34%. This is comparable with the data in northern China (29 ± 22%) [41] and further validates that a large amount of the PM_2.5_ in Shanghai is from the northern region [5,22,26,32].

Ozone is another major pollutant in Shanghai [12]. It also varies with the season and with meteorology [5,13,16]. The peak of its concentration occurs in summer due to higher temperatures and more sunshine [5]. Although the concentration of O_3_ has been recognised by the municipal government in Shanghai since 2016, the complexity in the formation of O_3_ makes it hard to regulate its emission [5,42]. In 2016, concentration of O_3_ accounted for 57.8% of primary pollutants in summer, and also indicated that a steady state level had formed in Chongming. [10]. However, O_3_ greatly increased in 2020, with an opposite trend to NO_2_, because of the complex oxidation effect of O_3_ with NOx [5,26,41,42]_._ The average increase (7%) of O_3_ in our study was in accordance with that at the non-roadside sites [5]. As our data were all from non-roadside sites, O_3_ increased before the lockdown period except for in Chongming, which has large areas of green plants. This validated the decrease in NO_2_ (*p* < 0.001) and that VOCs could increase O_3_ [41], and also indicates that a balance has been reached in Chongming. Concentrations (16%) during the lockdown period were comparable with those at the roadside sites (17%) in Shanghai [5]; and increased more than 2.5 times those in northern China [33,41]. All five major pollutants and the AIQ varied much more than the averages all over China [34].

### 4.1. Major Pollutants in Different Functional Areas Duiring COVID-19

To our knowledge, most research stresses the effect of lockdown on some of the airborne pollutants. Comparatively, fewer studies trace pollutant changes over long periods of time and compare data between pre-lockdown and post-lockdown [2,9,35,43]. They relate empirical evidences to their observed major pollutants and obtain results of temporary reductions in these pollutions during the lockdown period [2,9,35]. Although most parts of the world still suffer from COVID-19 and its variants, with more than 27 million confirmed cases as of 4 June 2021 [1], few comprehensive analyses are carried out to compare pre- and post-lockdown’s impact on pollutants [4,5]. As the first reported COVID-19 case was in China, which did well in fighting against the virus, this offers an excellent opportunity to study the air quality before, during and after the blockade [6,44].

Concentrations of these pollutants were significantly correlated with weather conditions [45,46,47]. The average temperature was about 8 to 9 °C with relatively high pressure in Shanghai in January. These conditions aggravated air pollution (Meteorological Administration Official Website). Furthermore, sand from the Mongolian desert was carried to Shanghai in winter. This increased the PM_10_ concentration in the air [10]. If the sand dust superposed the impact of local high pressure, it would spread further, from the north to the south Yangtze Delta area, including Shanghai, to increase the concentration of PM_2.5_ there [5,25,45,47]. Rainfall and wind speed also matter [5,45]. The average precipitation is 116.06 mm and 69.25 mm in January and February, respectively [7]. The rainfall is much higher than on the same dates in the years 2018 and 2019. This indicates that there was enough rain over the lockdown period in year 2020. The rain also lowered pollutant concentrations [16,22,29]. The average wind speed was 3.75 m/s, a low speed for the spread of airborne pollutants in Shanghai [10]. Thus, although there was enough rainfall, the low wind speed and high pressure still increased airborne pollutants. The meteorology did influence the concentrations of these pollutants, however, the effects are different for different types of pollutants. Rainfall and wind towards the open sea reduced most of the pollutants. The blockade also had an effect on the pollutants. The shutdown of most heavy industry, limited traffic and prohibited parties all contributed to a reduction in most pollutants.

### 4.2. Availability of Lockdown Policy’s Response to the Diffusion of COVID-19

As we know, the negative effects of COVID-19 and its variants are not only related to human health [18,20,29,47], but are also socio-economic [18,29]. The lockdown policy indirectly offered a chance to study and compare environmental changes in China before, during and after the lockdown period [1]. Policies like locking down cities and shutting down factories could reduce the emission of air pollutants from transport and production [3,16,18,23]. Xuhui, Chongming and Jinshan, with their various natural landscapes and anthropologic facilities, are different functional areas in Shanghai. Comparing pollutant changes in the three areas can enable discussions on how the pollutant concentration was affected by lockdown itself and other determinants [3,16,23,29].

As mentioned earlier, lockdown included both the restriction on nonessential travel and the interruption of manufacturing activities [3,26,43]. Given that the relatively shorter lockdown in Shanghai during the Spring Festival, i.e., 24 January to 9 February, was largely targeted to traffic, Table 1 shows the traffic control in Shanghai as a background applicable to all three sites. The results are in accordance with those in Figure 2, i.e., only two or three small peaks were found for SO_2_, NO_2_ and PM_10_, the main resource of which is fuel combustion [9,25,38]. The general increase during this period is in accordance with the “holiday effect” [46]. The small pollutant concentration peaks were mainly from cooking and traffic, whereas low concentrations were found in both O_3_ and PM_2.5_. As mentioned above (Figure 3 and Figure 4), O_3_ levels were reduced because of the increase in NO_2_ and the organic reduction produced by human activities [42,48]. The decreasing rate before lockdown (3%) was almost the same as the value during the lockdown period in northern China [41], whereas the source of PM_2.5_ is external and has less correlation with the local human activities in Shanghai [25]. Only SO_2_ has significant correlations with all other pollutants, because it is mainly from fuel combustion and can attach to PM_2.5_ and PM_10_ [22].

The global air quality experienced a dramatic improvement throughout the pandemic of COVID-19 [34,46] because the lockdown supplied a relatively non-industrial period [34,35,36,37,38,39,40,41,42,43,46,47,49,50]. Isolation is an effective way to block the spread of COVID-19 [34,35,36,37,38,39,40,41,42,43,44,45,46,47]; however, this means that people have to be restricted to their homes or somewhere else. They are unable to go out for work, and plants have to be stopped or reduce their production. This results in a lack of living material in some places, and could increase human’s mental problems, such as anxiety and depression [20,47], which would increase the instability of society [20]. The progress of the social economy is largely dependent on the development of heavy industries, which discharge greenhouse gases and accelerate global warming [2].

### 4.3. Implication of Environment Policies Taken by China

The lockdown period in China during COVID-19 provides an effective model that could be used throughout the whole world. The concentration of NO_2_ in oceans was three to four times lower than those in total land, without the Antarctic [41], which has a huge ozone hole and a low NO_2_ concentration [42]. The data from Shanghai highlight efforts of keeping pollutant concentration low for years when its growth trajectory takes sustainability into account [17,22]. Although the improvement in air quality is good within the lockdown period, it is unlikely to be maintained for long if the economy is still supposed to grow at a high speed. Necessary policies should be made to maintain or even improve the post-lockdown air quality [43]. Wiser policies than the forced reduction in industry and transport should be taken into consideration [23]. Green energy sources such as wind, water and photovoltaic power generation are good alternatives to traditional fossil consumption, which is high in pollution. Additionally, China’s early response to the concept and task of “carbon neutralization” and “carbon peak” will help to better safeguard the world’s environment and stabilize the climate. Furthermore, the people in China have quickly and positively responded to the Blue Carbon Initiative. The plan’s aim is to ensure economic growth on the basis of recovering a healthy, natural environment. It is a good way to control carbon’s release into the atmosphere [48,49].

### 4.4. Other Positive Effects Brought by the Lockdown Policy

New destructive variants could be found in the future because of the rapidly mutating nature of the virus [2]. There have already been many types of variants, especially *δ*(plus)- and *λ*-variants, with greater infection and spread speeds. This not only puts Europe back on “war footing” [45], but also causes serious effects in Fujian, China. Fujian has completed the third round of virus detection and cancelled the Mid-Autumn festival holiday in many places. Zhongshan Hospital, affiliated with Xiamen University, implemented closed management and the Tong‘an District has designated several closed control areas to implement regional closures, keep people indoors and close factories [46]. Lockdown policy is proven to be effective in blocking the spread of the virus [1,6,7,45] through limiting traffic and the number of factories and parties [9,36,41]. Besides effectively improving air quality [9,10,11,19,20,21,24,30,31,34,38,45,46,47], people have formed the habit of wearing masks, using dining tables, disinfecting upon leaving or entering their homes, timely nucleic acid detection, and closing factories that produce non-essential goods [33]. People also tried to use the minimum of living supplies during the outbreak, and put more energy into the pursuit of spiritual well-being [27,35,50]. Generally, life is lived realistically with controlled material needs [36,50]. In addition, in the face of the epidemic, people rethink the value of life, cherish their work, promote harmony in economic growth and clean environments, and stabilize societies.

## 5. Conclusions

There is no doubt that COVID-19 and its variants bring serious negative impacts to the world economy and people’s lives and health. However, China’s lockdown policy has not only stopped the spread of the viruses, but also provided scientists with an opportunity to study how pollutants change with less human intervention. This paper studied changes in five major airborne pollutants in different functional areas of Shanghai to evaluate the impact of lockdown policy on air quality. Overall, the COVID-19 lockdown has changed the five major pollutant concentrations, in both the long and short terms, at all three functional areas. Concentrations of all five pollutants, except for earth surface O_3_, decreased. SO_2_ had significant correlations with all other pollutants. PM_2.5_ and PM_10_ are more externally input than locally produced. NO_2_, SO_2_ and PM levels were sharply reduced in Jinshan and Xuhui due to the limited usage of fossil fuel in heavy industry and traffic. All these activities could meet the necessary needs of people and help to maintain better air quality. Besides the environmental improvement, COVID-19 lockdown gave people a chance to rethink the value of life, cherish their jobs, and maintain the harmony of a clean environment and the progress of the economy. All of this is helpful to establish sustainable societies.

## Figures and Tables

**Figure 1 ijerph-18-10613-f001:**
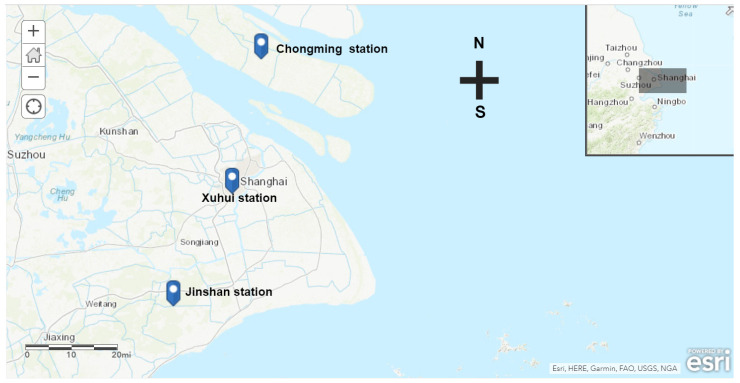
Sampling areas in Shanghai, China.

**Figure 2 ijerph-18-10613-f002:**
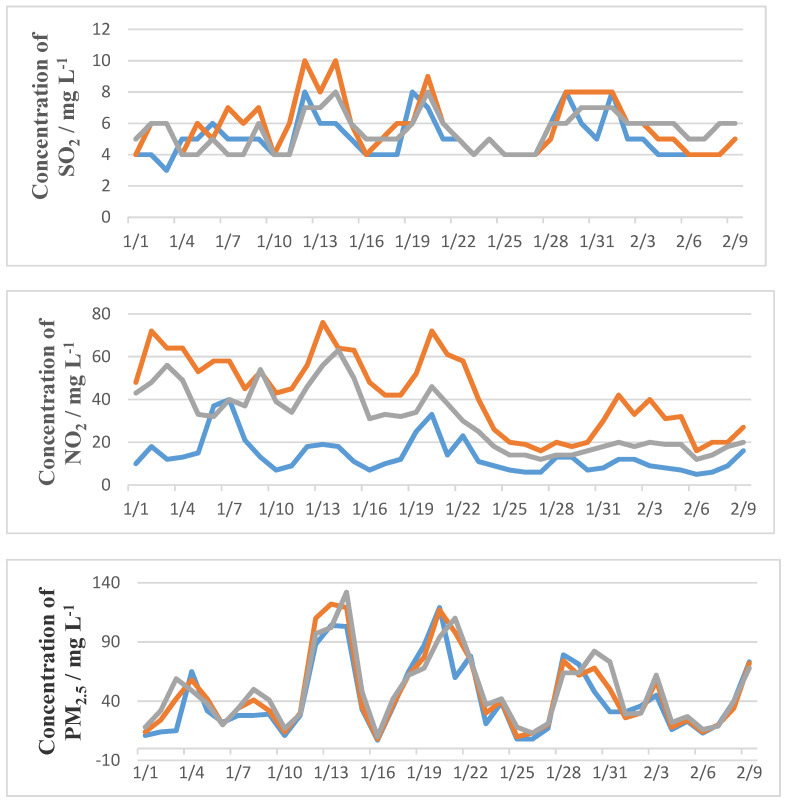
Changes in major pollutants from 1 January to 9 February with lockdown beginning on 23 January in different parts of Shanghai.

**Figure 3 ijerph-18-10613-f003:**
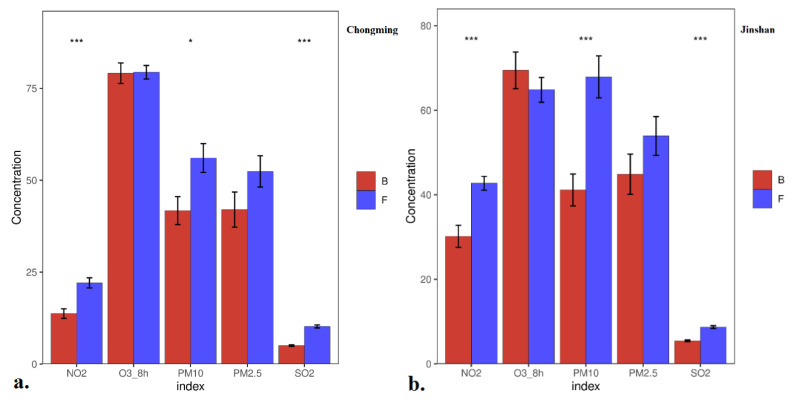
One-way ANOVA analysis for different parts and for all of Shanghai before and during COVID-19: during (B) and before (F) lockdown period. (**a**–**d**) represents concentrations of different pollutants, i.e., NO_2_, O_3_, PM_10_, PM_2.5_ and SO_2_ in Chongming, Jinshan, Xuhui as different functional areas and average values from the three functional areas of Shanghai.

**Figure 4 ijerph-18-10613-f004:**
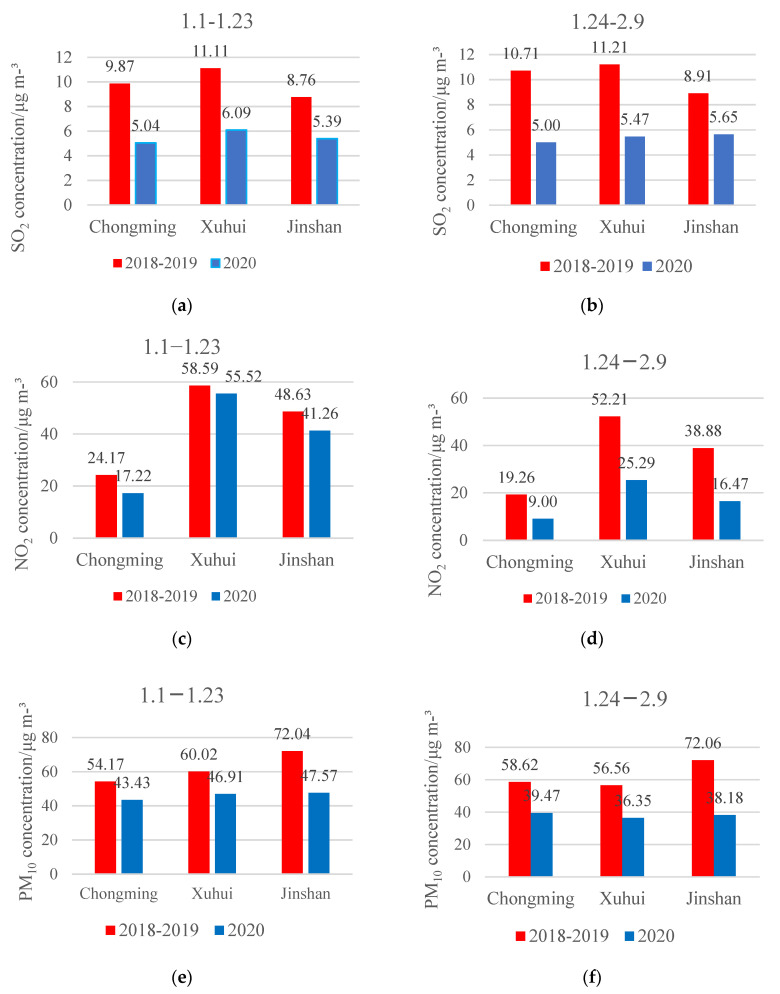
Average concentrations of different pollutants before (1.1–1.23) and during COVID-19 lockdown (1.24–2.9) in different years in Shanghai: The data on each panel are the average concentrations of each pollutant on the corresponding date. (**a**–**j**) are respective comparison of SO_2_, NO_2_, PM_2.5_, PM_10_ abd O_3_.

**Table 1 ijerph-18-10613-t001:** Traffic control in January and February of 2020 in Shanghai from Shanghai Bureau of Statistics.

Transportation by Different Ways	January	February	Differences
Total amount of cargo transported (10,000 tons)	10173.28	8583.22	−8.48%
Highway	3560	2200	−23.61%
Airport	30.22	19.16	−22.40%
Railway	1050.49	68.25	−87.80%
Harbour	6.3	0.01	−99.68%

## Data Availability

Not applicable.

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
