# Peer review of "Reflections Based on Pollution Changes Brought by COVID-19 Lockdown in Shanghai"

_ijerph, 2021, doi:10.3390/ijerph182010613_

Round 1
Reviewer 1 Report
Overall, this study is interesting and showed the pollutant concentration changes in three parts of Shanghai, corresponding to three emission sectors, during the COVID-19 lockdowns and compared to results before the lockdown and in 2018-2019. However, this manuscript seems to be prepared in rush. Too many types in the text. Terms are not consistent. Figure panels are not well described. I think this manuscript must be significantly edited and the discussion should be improved before considering publication. I have specific comments followed by minor issues I caught.
Introduction: I think the introduction can be improved. There should be some sentences on the common sources of the pollutants you studied here and relationship between emissions and pollutant concentrations. The typical reductions of pollutants in other cities reported by other papers.
Section 1.2: The two equations in Equation (1) should be two separate equations. Equation (2) and (3) are not equations. They should be in the text right after the two equations. Please define concentration breakpoint.
In section 1.1 or Section 1.2, authors should clearly write out the dates of the start and end of the lockdown. It is easier for later on discussion and no need to repeat in the figure captions.
Section 2.1. when you discuss the chemistry of NOx, O3, VOCs, you may want to refer to this study as an overall thought:
https://www.nature.com/articles/s41557-020-0535-z
When discussing this, you may also want to check a few more literatures about interesting observations, elevated O3 and PM in China during the COVID-19 lockdowns. These discussions will help you relate your results to other cities and bigger context.
Line 212-215 and line 49-50: There are definitely studies comparing air pollutants concentration levels during, before, and after the lockdowns, and comparison to previous years of the same periods. These studies include surface measurements of NOx, CO, SO2, O3, PM, as well as remote sensing observations of NO2 columns, CO columns, and aerosol AOD over different urban regions in the world. Actually several papers are on the Chinese cities and industrial regions, including both satellite observations and surface observations. I understand it is hard to follow all the hundreds of publications over only 1.5 years on this topic, but it is unfair to say “there is almost no studies…”. I strongly suggest the authors reading and citing a few more representative works on this topic and rephrase those sentences.
Line 229: about the rain fall in Jan and Feb 2020. How do they compared to 2018 and 2019. Since you compared the pollutants concentrations in three years, you may want to at least mention the precipitation comparisons in three years.
Second half of section 4.1: what is your conclusion? Does meteorology matter? Are those changes or decreased you observed in three parts of Shanghai were caused/contributed by the COVID-19 lockdown or meteorology changes? Can you estimate the impact of the change of meteorology parameters on the pollutant concentrations and concentration changes you observe, if there is any changes in meteorology parameters? If you cannot quantify the impact, at least discuss it briefly and refer to some relevant method.
Line 283-289: In this study there is no CO, CO2, CH4 or other greenhouse gases results. Therefore, I think this part of the text is off the topic of this study.
Conclusion: Line 301-308 should be deleted. The sentence in 310-311 should be deleted. Need to rewrite. Think bout summarizing your direct results and concise implications.
Figure 2: can be improved. The legend of the three parts must be bigger and you don’t have to repeat the legend for every pollutant. The date labels are too small to read. Maybe only label every three days and make the font bigger.
Minor issues:
Figure 4: which panels are “before”, which ones are “during”? what are the numbers on top of each panel?
Please be consistent as COVID-19, not Covid-19, through the text.
There are many errors or typos, and I only caught some of them as listed below:
Line 11: airborne pollutants
Line 14: PM2.5 and PM10
Line 18: societies.
Please check the last keyword.
Line 30: suggest rewording: “The COVID-19 lockdowns also greatly impacted air quality.”
Line 31: during what?
Line 38: harmful, human
Line 42: pollutants
Line 45: “In addition,” is better than “Besides,”
Line 50: “during” the lockdowns. “period” can be deleted.
Line 52: economic. People’s lives
Line 60: , and daily average concentrations were used.
Line 61: Three regions in Shanghai, i.e., …
Line 69: replace “exhale” with “emit”.
Line 71: collected an
Line 72: 613 values during 2018-2020…
Line 77; you meant “lockdown”?
Line 87: concentrations
Line 91 negative
Line 92: after the lockdown started as it is mainly…
Line 96: peaks
Section 2.2: increased or decreased by x%. There should be “by” before the percentage results.
Line 124: no significant effect on either PM2.5 or O3 was…
Line 128: fossil fuel combustion
Line 144-147: decreasing rates were x% per year or? The unit of the decreasing rate is not correct.
Line 207: and also indicated that the steady state has formed in Chongming.
Line 231: “absorb” is not the correct term here. I think here you meant “scavenge” the pollutants or wet deposition of pollutants.
Line 250: peaks!!!
Line 261: global
Line 264: restricted at home
Author Response
Overall, this study is interesting and showed the pollutant concentration changes in three parts of Shanghai, corresponding to three emission sectors, during the COVID-19 lockdowns and compared to results before the lockdown and in 2018-2019. However, this manuscript seems to be prepared in rush. Too many types in the text. Terms are not consistent. Figure panels are not well described. I think this manuscript must be significantly edited and the discussion should be improved before considering publication. I have specific comments followed by minor issues I caught.
Reply: Yes! Thanks for your kindly advice. We also fund this after submitted the paper. And this time, after revised the minor issues that you and the other Reviewer suggested. We make a thoroughly modification, we carefully revised the Resutls and Discussion.
Introduction: I think the introduction can be improved. There should be some sentences on the common sources of the pollutants you studied here and relationship between emissions and pollutant concentrations. The typical reductions of pollutants in other cities reported by other papers.
Reply: Yes. We modified the Introduction and especially varied the last graph. It is as follows:
There have been a lot of studies on enviromental changes during the breakout and lockdown of Covid-19[1,4-6,13,23,24,36,41-45,55]. However, almost all studies concentrate on the lockdown period[5,6,40-55] without comparison among pre- during and after lockades. Besides, seldom studies refer to functional areas [5,24]. Covid-19 and its variants alter the world from many aspects, including both social and natural enviroments, and globle econony. Covid-19 also have many other effect on humen’s life, physical and mental health[1,3,20,23,35]. However, it also provides a special chance to hunman to rethink the value of life, making appropriate measures to balance between ecomomic growth and clean natural environment[4,11]. So, although most effects brought by the variants are negtive, they give mankind the chance to restart the world and this is helpful to establish stainable societies.
Section 1.2: The two equations in Equation (1) should be two separate equations. Equation (2) and (3) are not equations. They should be in the text right after the two equations. Please define concentration breakpoint.
Reply: Equation (1) is seperated into two equations now. And the orignal Equation (2) and (3) is deleted. Now they are rewritten and just after the two equations.
In section 1.1 or Section 1.2, authors should clearly write out the dates of the start and end of the lockdown. It is easier for later on discussion and no need to repeat in the figure captions.
Reply: Thanks for your advice. Now a presentation is added at beginning of section 1.2, it is as “The lockade period was from 23rd Jan to 9th Feb.”.
Section 2.1. when you discuss the chemistry of NOx, O3, VOCs, you may want to refer to this study as an overall thought: https://www.nature.com/articles/s41557-020-0535-z. When discussing this, you may also want to check a few more literatures about interesting observations, elevated O3 and PM in China during the COVID-19 lockdowns. These discussions will help you relate your results to other cities and bigger context.
Reply: Thanks for telling. Yes, this paper is much more comprehensive. And this time we have use this reference (21) insteading the original one.
Line 212-215 and line 49-50: There are definitely studies comparing air pollutants concentration levels during, before, and after the lockdowns, and comparison to previous years of the same periods. These studies include surface measurements of NOx, CO, SO2, O3, PM, as well as remote sensing observations of NO2 columns, CO columns, and aerosol AOD over different urban regions in the world. Actually several papers are on the Chinese cities and industrial regions, including both satellite observations and surface observations. I understand it is hard to follow all the hundreds of publications over only 1.5 years on this topic, but it is unfair to say “there is almost no studies…”. I strongly suggest the authors reading and citing a few more representative works on thits topic and rephrase those sentences.
Reply: Thanks for telling. The presentation in line 49-50 has been changed to “However, almost of theses studies concentrate on the lockdown period [5,6,40-55] without comparison pre- and after lockades.” And the presentation in line 212-215 has been changed to “To our knowledge, most work stress the effect of lockdown on some of the airborne pollutant. Comparatively, less studies to trace pol-lutant changes in long time series and comparing data between pre-lockdown and post-lockdown[2,9,35,45].” Hopefully, the presentation is appropariate this time.
Line 229: about the rain fall in Jan and Feb 2020. How do they compared to 2018 and 2019. Since you compared the pollutants concentrations in three years, you may want to at least mention the precipitation comparisons in three years.
Reply: Yes. We have checked the rainfull at the same data in year of 2018 and 2019. And we find that the rainful is much less than those in 2020. It is not suprising as the winter is warmming and warming and extreme weather conditions are more and more. Compared with the temperature in both years, the temperature in 2020 is much higher than those in years of 2018 and 2019. So, the presentation now is changed to “The rainfall is much higher than those at the same data of year 2018 and 2019. This indicates there was enough rain over the lockdown period in year 2021.” In line 231-233 this time. Hopefully it is much better. Thank you!
Second half of section 4.1: what is your conclusion? Does meteorology matter? Are those changes or decreased you observed in three parts of Shanghai were caused/contributed by the COVID-19 lockdown or meteorology changes? Can you estimate the impact of the change of meteorology parameters on the pollutant concentrations and concentration changes you observe, if there is any changes in meteorology parameters? If you cannot quantify the impact, at least discuss it briefly and refer to some relevant method.
Reply: Thanks for your reminding. Now we added some discusion at the end of the seconde paragraph. They are as “The meteorology did influence the concentrations of these pollutants. However, the effects are different to different types of pollutants. Raimfall and wind towards the open sea deduce most the pollutants. Blockade also matter the pollutants. Shutdowned most heavey industry, limited traffic, and prohibited parties all deduced most pollutants.”
Line 283-289: In this study there is no CO, CO2, CH4 or other greenhouse gases results. Therefore, I think this part of the text is off the topic of this study.
Reply: Yes. The three greenhouse gases is deleted this time. Now in line 293-295.
Conclusion: Line 301-308 should be deleted. The sentence in 310-311 should be deleted. Need to rewrite. Think bout summarizing your direct results and concise implications.
Reply: Yes. Your are right. We wrote this part. And now the whole presentation is as “There is no doubt that COVID-19 and its variants bring serious negative impact on the world economy and people's lives and health. However, China's lockdown policy has not only stopped the spread of the viruses, but also provided scientists with an opportunity to study how pollutants change with less human intervention. This paper studies changes of 5 major airborne pollutants in different functional areas of Shanghai to evaluate the impact of lockdown policy on air qualities. Overall, the Covid-19 lockdown has changed the 5 major pollutant concentrations in both long and short terms at all the three functional areas. All the five concentrations except for earth surface O3 decreased. SO2 had significant correlations with all other pollutants. PM2.5 nd PM10 are more external input than local procuced. NO2, SO2 and PM matters sharply reduced in Jinshan and Xuhui dued to limited usage of fossil fuel in heavy industry amd traffic. All these activies could meet humen’s neccesary needs and keep a better air quality. Besides the enviromental improvement, Covid-19 lockdown gave peple chance to rethink the value of life, cherish their jobs, keep hamony of clean environment and progress of economy. This is helpful to establish sustainable socities.” We think this time it much more focuses on the study aims. Thanks for reminding.
Figure 2: can be improved. The legend of the three parts must be bigger and you don’t have to repeat the legend for every pollutant. The date labels are too small to read. Maybe only label every three days and make the font bigger.
Reply: We replotted this figure with larger ledend and data labels. We also label the data every three days. And only supply one lable for all the funtional areas. The figures are as follows:
Figure 2. Changes of major pollutants from 1st Jan to 9th Feb with lockdown beginning date of 23rd January at different parts of Shanghai.
Minor issues:
Figure 4: which panels are “before”, which ones are “during”? what are the numbers on top of each panel?
Reply: The label of fig.4 is now changed to “Average concentrations of different pollutants before (1.1-1.23) and during (1.24-2.9) COVID-19 in different years in Shanghai: The data on each panel is the average concentration of each pollutant on the corresponding date.”
Please be consistent as COVID-19, not Covid-19, through the text.
Reply: Thanks for telling. We have changed all the Covid-19 to COVID-19 in the Ms.
There are many errors or typos, and I only caught some of them as listed below:
Line 11: airborne pollutants; Line 14: PM2.5 and PM10; Line 18: societies; Please check the last keyword. Line 30: suggest rewording: “The COVID-19 lockdowns also greatly impacted air quality.” Line 31: during what? Line 38: harmful, human; Line 42: pollutants; Line 45: “In addition,” is better than “Besides,” Line 50: “during” the lockdowns. “period” can be deleted. Line 52: economic. People’s lives. Line 60: and daily average concentrations were used. Line 61: Three regions in Shanghai, i.e., …. Line 69: replace “exhale” with “emit”. Line 71: collected an. Line 72: 613 values during 2018-2020… Line 77; you meant “lockdown”? Line 87: concentrations; Line 91 negative; Line 92: after the lockdown started as it is mainly…; Line 96: peaks. Section 2.2: increased or decreased by x%. There should be “by” before the percentage results. Line 124: no significant effect on either PM2.5 or O3 was…; Line 128: fossil fuel combustion. Line 144-147: decreasing rates were x% per year or? The unit of the decreasing rate is not correct. Line 207: and also indicated that the steady state has formed in Chongming. Line 231: “absorb” is not the correct term here. I think here you meant “scavenge” the pollutants or wet deposition of pollutants. Line 250: peaks!!! Line 261: global; Line 264: restricted at home.
Reply: This time we have tried our best to correct the errors according to your kindly advice. Thanks again for your careful check.
Best regards!
Submission Date
14 September 2021

Reviewer 2 Report
The content of the submitted manuscript is good but the presentation way of current form is not fulfilling the journal requirements. Modification is needed to consider for publication.
Figure
Figure quality is not up to the mark. So, I request you please update the quality of each and every figure, if required.
- Title of the paper
The title of the paper looks good but in the same time, it can be modified to represent the manuscript in a better way.
Abstract
- The abstract is not well written
- You should include some of the main finding in the abstract section.
Abstract should have a conclusion of the study.
Introduction
- The objective of the study is also not clearly mention.
- Add more on the basic of the problem in the introduction
- More details about COVID-19 are required in the introduction section.
- The author should focus mainly on the importance and significance of the study.
- I suggest the author to demonstrate what does the paper add to the current literature? and what new knowledge is added by this study?
Literature review part is need to be updated.
Include a table Like
|
Study Area (place) |
Pollutant Types |
Key Observations |
Author (year) |
|
|
|
|
|
|
|
|
|
|
|
|
|
|
|
|
|
|
|
|
Add the unique of this study compared to other studies discuss the same issue.
-Discus merits and limitations of technique applied.
Material and Methods
- The material and method section is too weak in the manuscript and you need to focus on it more.
Result and discussion
- The presentation fails to discuss the summary, and trying to some of vague reason which is not the explanation.
- The explanation for the critical analysis is not sufficient, although some of the good points have has been identified.
Conclusion
- Please rewrite the conclusion with the proper explanation in the R & D.
References
Reference section should be increased with number of recent studies.
e.g. COVID – 19: Air pollution remains low as people stay at home. Air Quality Atmosphere and Health DOI: 10.1007/s11869-020-00842-6
Other comments:
English editing is needed in some parts of the manuscript.
Abbreviations should be explained before the introduction.
Author Response
Reply to Comments and Suggestions for Authors
The content of the submitted manuscript is good but the presentation way of current form is not fulfilling the journal requirements. Modification is needed to consider for publication.
Reply: Thanks for your kindly recommendation and we have modificated the paper this time as best as we cam.
- Figure
Figure quality is not up to the mark. So, I request you please update the quality of each and every figure, if required.
Reply: We have reploted Figure 2 and tried our best to supply figures with enogh quallity.
- Title of the paper
The title of the paper looks good but in the same time, it can be modified to represent the manuscript in a better way.
Yes, you are right. Now we have changed title of the paper to “Reflections based on pollution changes brought by COVID-19 lockdown in Shanghai”. It is much comprehensive the previous. Thanks for telling.
- Abstract
- The abstract is not well written; -You should include some of the main finding in the abstract section. Abstract should have a conclusion of the study.
Reply: We compeletly rewrote the abstract as is: COVID-19 and its variants has been changing the world. The spread of variants brings severe effects on global economy, humen’s life and health, as well as society steady. Lockdown is proved to be effective in stopping the spread.It also provide a chance to study natrual enviromental changes with human’s small interfernces. This paper just rightly aims to evaluate the impact of lockdown on 5 major airborne pollutants, i.e., NO2, SO2, O3, PM2.5 nd PM10, in 3 different functional regions, i.e., Chongming, Xuhui and Jinshan of Shanghai. Changes of the same pollutants from the 3 regions in the same/different periods were all studied and compared. Overall, the COVID-19 lockdown has changed pollutant concentrations in long and short terms. Concentrations of 4 pollutants decreased, except for those of earth surface O3increased. SO2 had significant correlations with all other pollutants. PM2.5 and PM10 are more external input than local procuced. NO2, SO2 and PM matters sharply reduced in Jinshan and Xuhui dued to limited usage of fossil fuel. Lockdown improved the air quality. People get chance to rethink the value of life, harmony between economic progress and enviromental protection. This is helpful to establish sustainable societies.
- Introduction
The objective of the study is also not clearly mention. Add more on the basic of the problem in the introduction. More details about COVID-19 are required in the introduction section. The author should focus mainly on the importance and significance of the study. I suggest the author to demonstrate what does the paper add to the current literature? and what new knowledge is added by this study? Literature review part is need to be updated. Include a table Like. Study Area (place). Pollutant Types. Key Observations; Author (year). Add the unique of this study compared to other studies discuss the same issue. -Discus merits and limitations of technique applied.
Reply: We wrote this part along with addition and changes some reference. Of cource, the data were update to 15th September with addition of some basic problems, which makes background of COVID-19 more abundant. We stress our study domain at the beginning of the second paragraph of Introduction. It is as “The development of human society relies too much on traditional energy sources, such as coal and oil, which produce large quantities of polluting gases. So, air pollution is a major problem at present, which has serious impacts on climate change and human health. We need to know the basic change of pollution gases with the minimum human intervention, so as to compare with the pollution gases produced by the normal operation of human society and obtain the basic data of human intervention. The COVID-19 lockdown just rightly give human the chance[9-11]. Different regions in the world had different polluting gasses [9-39,41,42,44].” We did not put a table here because we think the addtion of table here breaks the integrality of the manuscript. And there are comparison of pollutants in different regions in the discussion part. So, we only list the references here without overelaboration. And we rewrote the beginning of the last paragraph of this part as “There have been a lot of studies on enviromental changes during the breakout and lockdown of COVID-19[1,10,13,15,20,21,27,31-36]. However, most of theses studies concentrate on during the lockdown [10,11,19,27,31-34,36-44] without comparison among pre- and after lockades. Besides, seldom studies refer to functional areas [10,20]. The two issues were exactly focused on by our study.” The discussion of the methods used in this ms in in the Methods as “One-way ANOVA is used to compare the concentration of the same pollutants bettween different groups, including pre- and during the lockdown in year of 2020, and between years of 2018-2019 and 2020. NMDS is a data analysis method that simplifies the research objects (samples or variables) in multidimensional space to low-dimensional space for positioning, analysis and classification, while preserving the original relationship between objects. According to the pollutant information contained in the samples, it is reflected in the multidimensional space in the form of points, and the degree of difference between different samples is reflected by the distance between points, and finally the spatial location map of the samples is obtained. Cluster analysis is a method to simplify data through data modeling. It is a set of statistical analysis techniques that divide research objects into relatively homogeneous groups. Clustering analysis is also called classification analysis or numerical classification. Cluster analysis can start from the sample data to automatic classification. If the variables are independent of each other, it can quickly process large data sets with automatically determine the optimal classification number of these classification and continuous variables.”. Hopefully, we state the problems clearly this time. Thanks for your advices.
- Material and Methods
-The material and method section is too weak in the manuscript and you need to focus on it more.
Reply: Yes, as we said above, we have added a paragraph at the end of the section to explain the characteristics of the methods including One-way ANOVA, NMDS and cluster analysis.
- Result and discussion
The presentation fails to discuss the summary, and trying to some of vague reason which is not the explanation. The explanation for the critical analysis is not sufficient, although some of the good points have has been identified.
Reply: The presentation is rewritten this time to put in many essencial discussion on the spread of COVID-19 and its variants as well as the advantages of lockdown poly. Some references are the newest reports. Please see the text for detail.
- Conclusion
- Please rewrite the conclusion with the proper explanation in the R & D.
Reply: This paprt has been rewritten.
- References
Reference section should be increased with number of recent studies.
e.g. COVID – 19: Air pollution remains low as people stay at home. Air Quality Atmosphere and Health DOI: 10.1007/s11869-020-00842-6
Reply: We cite this reference as you suggested. And had cited more references to state our opinon more clearly. Thanks four telling.
- Other comments:
English editing is needed in some parts of the manuscript. Abbreviations should be explained before the introduction.
Reply: We have tried our best to modify the spelling and grammar mistakes as kindly suggested by both Reviewers. As to the abbrevisions in the abbstract, do you mean COVID-19, NO2, SO2, O3, PM2.5 and PM10? We think they're all stereotypes that need no explanation. Thank you!
Best Regards!
Fang Zhang on behalf of all authors

Round 2
Reviewer 1 Report
Hello,
I have gone through both your cover letter and revised manuscript. There are still a few things you need to check and correct before it can be published. For example, equations are still NOT revised, fonts in Figure 3 are too small to read, and numbers labelled in Figure 4 need to check.
I still found some typos or inconsistency in your revised manuscript, marked in the attachment. please address them.

Author Response
Reply to Reviewer#1 ‘s coments: point to point:
- Figure 3 is replotted with bigger fonts.
- Numbers labelled in Figure 4 now is corrected.
- "Rate" is deleted at line 180.
- We have checked Figure 4, and we have found no errors with the same values. Maybe you see an old one? Please check. Thank you very much!
- We checked the diameter of the three areas: 3 average data, which is an increase, not a larger decrease. So we just change "increasing rate" to "increase" in line 237.
- “pollution of PM10” is corrected to “pollution of PM10” in line 261.
- Yes, It is 2020 rather than 2021 in line 267. Thanks for your careful examination.
- “deduce” is changed to “reduce” in line 272.
- "were" is corrected in line 293. Thanks again.
- The word "COVID-19" in line 348 has now been deleted.
- Change "local" to "locally" in line 357.
- We replace "human’s necessary needs " with "necessary needs of people" in line 259.
- We tried our best to check the words spelling and hope that there is no errors this time.
We really read a lot from you and thank you for your care and support in reading our paper.
Regards!
Fang Zhang on behalf of all authors
